# EARTH: Epidemiology-Aware Neural ODE with Continuous Disease Transmission Graph

Guancheng Wan[1 2]  Zewen Liu[1]  Xiaojun Shan[3]  Max S. Y. Lau[4]  B. Aditya Prakash[5]  Wei Jin[1]

## Abstract

Effective epidemic forecasting is critical for public health strategies and efficient medical resource allocation, especially in the face of rapidly spreading infectious diseases. However, existing deep-learning methods often overlook the dynamic nature of epidemics and fail to account for the specific mechanisms of disease transmission. In response to these challenges, we introduce an innovative end-to-end framework called Epidemiology-Aware Neural ODE with Continuous Disease Transmission Graph (`EARTH`) in this paper. To learn continuous and regional disease transmission patterns, we first propose EANO, which seamlessly integrates the neural ODE approach with the epidemic mechanism, considering the complex spatial spread process during epidemic evolution. We also introduce GLTG to model global infection trends and leverage these signals to guide local transmission dynamically. To accommodate both the global coherence of epidemic trends and the local nuances of epidemic transmission patterns, we build a cross-attention approach to fuse the most meaningful information for forecasting. `EARTH` offers a more robust and flexible approach to understanding and predicting the spread of infectious diseases. Extensive experiments show `EARTH` superior performance in forecasting real-world epidemics compared to state-of-the-art methods. The code is available at https://github.com/GuanchengWan/EARTH.

[1]Department of Computer Science, Emory University, USA [2]Department of Computer Science, University of California, Los Angeles [3]Department of Electrical and Computer Engineering, University of California, San Diego [4]Department of Biostatistics and Bioinformatics, Emory University, USA [5]College of Computing, Georgia Institute of Technology, USA. Correspondence to: Wei Jin <wei.jin@emory.edu>.

*Proceedings of the 42nd International Conference on Machine Learning*, Vancouver, Canada. PMLR 267, 2025. Copyright 2025 by the author(s).

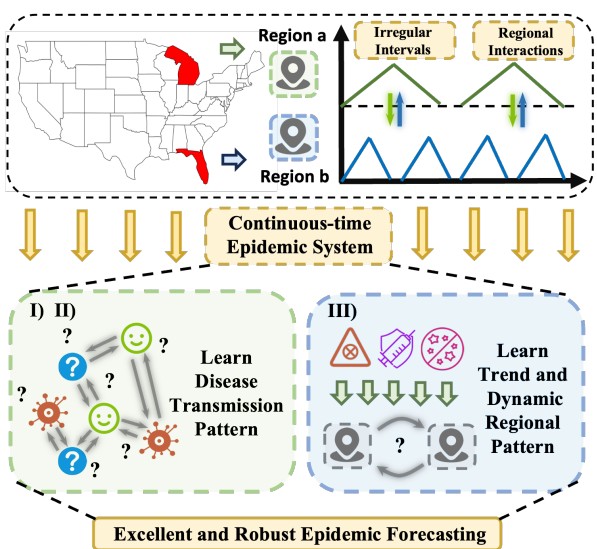

*Figure 1.* Problem illustration. Considering both *evolution of regional correlation signals* and *irregular sampling observation intervals* facts, we focus on the continuous-time epidemics. But existing solutions fail to **I)** learn disease transmission patterns with epidemic mechanism and **II)** address missing states. Additionally, they omit to **III)** learn global trends caused by external factors (*e.g.*, lockdowns) while developing dynamic regional transmission.

## 1. Introduction

The COVID-19 pandemic has resulted in millions of deaths and significant economic losses worldwide, severely disrupting social and economic systems (Martin et al., 2023; Pak et al., 2020). To address these challenges, there is a growing interest in epidemiological models, which are crucial for effective public health strategies and efficient medical resource allocation (Liu et al., 2024; Cm, 2020). Traditional models, such as the SIR model and its variants (Dehning et al., 2020), rely on mathematical differential equations to simulate disease spread but often depend on oversimplified assumptions (Funk et al., 2018; Kondratyev, 2013; Yang et al., 2023). To enhance performance, deep learning models like Neural Networks (Madden et al., 2024; Rodríguez et al., 2023) or Graph Neural Networks (GNNs) (Zhang et al., 2024e;b) have been explored. These models effectively represent interactions between entities (*e.g.*, regions) as graphs, capturing the spatial spread of disease through

message-passing mechanisms.

Nevertheless, given the dynamic nature of epidemic systems, existing work often neglects the *dynamic evolution of regional interactions*. For example, changes in people's behavior (*e.g.*, lockdowns) at a specific time step of one region, will greatly reduce the spread to surrounding regions in the following period. Existing efforts generally predict future epidemic profiles by modeling regional interactions from the whole series while overlooking these dynamic changes during the evolution. Furthermore, *irregular sampling observation intervals* are not considered. For instance, some regions may be unable to conduct routine reporting in the early stages of an epidemic due to limited resources. Current work simplifies this scenario by learning only regular intervals which is impractical in the real world. The problem illustration is detailed in Figure 1.

To tackle the aforementioned issues, neural ordinary differential equation (NODE) (Chen et al., 2018; Poli et al., 2021; Liu et al., 2025) stands out as a powerful approach to modeling the continuous-time system. Therefore, in this work, we take inspiration from NODE and focus on the *continuous-time epidemic system*, capturing the intricate dynamics more accurately. However, directly incorporating the neural ODE with the epidemic system faces nontrivial challenges. Firstly, it does not explicitly learn epidemic mechanisms and fails to provide insights to decision-makers. This motivates us to think: **I)** ***How can we generally combine the neural ODE approaches with epidemic mechanism?*** Some work (Arik et al., 2020; Mežnar et al., 2021) proposes hybrid models trying to combine the epidemic mechanism and deep learning methods. However, due to the limited observations of all disease states (*e.g.*, lacking data on susceptible individuals) in the real world, these models are not flexible and are unable to learn inherent epidemic mechanisms. In the meanwhile, some existing neural ODE approaches from other fields (Luo et al., 2023) fail in addressing this problem either. Thus, the following question naturally emerges: **II)** ***How can we learn continuous disease transmission under limited observations more flexibly?*** Apart from the aforementioned locally subtle spreading patterns, epidemics also exhibit a global infection trend. From a more macroscopic perspective, the infection trend can be seen as a longer-range and often multi-regional overall direction. This global signal impacts and changes the disease propagation, resulting in different spatial transmission patterns at different times. For instance, global political vaccination trends significantly alter local spatial transmission patterns. In regions with high vaccination rates, the spread of the epidemic slows down, and a "herd immunity" effect may even occur (Chauhan et al., 2023). However, previous work usually considers static geographic graphs or only learns the graph without accounting for the continuous evolution of global signals. This raises another intriguing question: **III)** ***How can we model the global infection trend and learn dynamic regional transmission patterns during continuous evolution?***

To address the identified questions, we propose an innovative and end-to-end framework for continuous-time epidemic modeling: **E**pidemiology-**A**wa**R**e ODE with Continuous Disease **T**ransmission Grap**H** (`EARTH`). To address question **I)** and facilitate epidemiology-informed transparency, we revisit the classic compartmental models (*i.e.*, SIR). In order to surpass previous efforts (Rodríguez et al., 2023) and fully leverage the expressive ability of neural networks, we propose a neural ODE-based Network SIR (Brede, 2012) to implicitly capture the continuous evolution of the regional propagation graph. To overcome the challenge **II)** we propose to initialize disease state features and feed them into a proposed **E**pidemic-**A**ware **N**eural **O**DE (EANO) module to learn inherent epidemic transmission pattern. Moreover, we attempt to achieve the target **III)**. We first obtain a long-range view of epidemic progression and establish a relationship with regions that share similar development patterns. To further consider dynamic regional transmission, we develop an innovative **G**lobal-guided **L**ocal **T**ransmission **G**raph (GLTG). Specifically, we fuse global trend indicative features for different regions with GNN. Then they are utilized to generate more fine-grained locally dynamic transmission graphs, which guide our EANO disease spreading during the evolution. Finally, we develop a cross-attention mechanism to accommodate both the global coherence of epidemic trends and the local nuances of disease transmission patterns. We conjecture that these two components together make `EARTH` a competitive method for epidemic forecasting.

Our principal contributions are summarized as follows:

❶ We are the first to harmonize the neural ODE with the epidemic mechanism, developing an innovative framework considering the time-continuous nature of epidemic dynamics while learning inherent disease spreading patterns.
❷ We further consider global epidemic trends and learn dynamic regional transmission patterns during continuous evolution within the end-to-end model.
❸ By integrating global coherence and local dynamics via a cross-attention mechanism, we achieve superior results on multiple epidemic forecasting datasets including COVID-19 and influenza-like illness.

## 2. Related Work

### 2.1. Epidemic Forecasting

Epidemic forecasting plays a crucial role in predicting the spread and impact of infectious diseases, enabling timely and effective public health interventions (Emanuel et al., 2020; Fine, 2015; Terris, 1993). Traditional models like the SIR (Susceptible-Infectious-Recovered) model (Hethcote, 2000) use differential equations to describe disease dynam-

ics but often rely on oversimplified assumptions (Dehning et al., 2020; Caals et al., 2017). Recent advances incorporate deep learning methods like Graph Neural Networks (GNNs) (Dai et al., 2022; Wan et al., 2024b), which better capture the complex interactions and spatial dependencies in disease spread from data-driven perspectives (Deng et al., 2020; Yu et al., 2023). However, these deep learning methods neglect the dynamic nature of the epidemic system. The issues of *regional correlation signals* and *irregular sampling observation intervals* remain unresolved, hindering the accurate capture of real-world epidemics. Therefore, in this work, we propose a general framework by seamlessly integrating epidemic mechanisms into Neural ODE, capturing the complex evolution of continuous-time epidemics.

## 2.2. Graph Neural Networks

Graph Neural Networks (GNNs) (Hamilton et al., 2017; Veličković et al., 2017; Huang et al., 2023; Wan et al., 2024a; Zhang et al., 2024a;d; Wan et al., 2025b) are widely recognized for processing non-Euclidean data structures, such as traffic networks (Wu et al., 2019). They update node representations by aggregating information from neighbors via message-passing (Zhang et al., 2024c; Zhang et al.; Chen et al., 2024a; 2023). Many studies have used GNNs for epidemic modeling (Sha et al., 2021; Wang et al., 2023), focusing on the spatial relationships in disease spread (Jhun, 2021; La Gatta et al., 2021), but often overlook dynamic transmission patterns. Our approach addresses this by concentrating on continuous-time epidemic modeling, using GNNs to integrate multi-region global trends and create disease transmission graphs for regional propagation.

## 2.3. Neural Ordinary Differential Equation

Neural ODEs extend discrete neural networks to continuous-time scenarios, offering superior performance and flexibility (Chen et al., 2018). They have been widely adopted in various fields such as traffic flow forecasting (Fang et al., 2021; Choi et al., 2021), continuous dynamical systems (Chen et al., 2024b; Huang et al.), and recommendations (Qin et al., 2024). Recent advancements have integrated GNNs with Neural ODEs, enhancing the modeling of complex dependencies in graph-structured data (Luo et al., 2023; Wan et al., 2025a). Some works have started to apply Neural ODEs to model classic epidemic spreading processes like SIR (Kosma et al., 2023). For a more comprehensive overview, readers are referred to the survey in (Liu et al., 2025). However, these approaches often rely on pre-defined epidemic models and do not fully capture the dynamic nature of regional interactions or global trends. In contrast to prior work, we extend this concept to investigate important continuous-time epidemic modeling by developing a framework that harmonizes the neural ODE with epidemic mechanisms while dynamically learning the

disease transmission graph. Since the epidemic system is time-varying, we first attempt to associate each region with the time-corresponding latent variable $\mathbf{Z}_v(t)$ by a parameterized ODE $\dot{\mathbf{Z}}_v(t) := d\mathbf{Z}_v(t)/dt = \psi_t\big(\theta_t; t; \mathbf{Z}_v(t)\big)$, which depicts the region-specific dynamic trajectory for series. Thus we can derive temporal dynamics at $T$ for all regions $\mathbf{Z}(T) \in \mathbb{R}^{N \times d}$ as follows:

$$\mathbf{Z}(T) = \mathbf{Z}(0) + \int_0^T \psi_t\big(\theta_t; t; \mathbf{Z}(t)\big)dt. \tag{1}$$

Here $\psi_t$ denotes the time manipulation function parameterized by $\theta_t$. In our framework, we leverage multi-layer perceptrons (MLP) for the time modeling module $\psi_t$ by default. Furthermore, for different regions, diverse and complex processes of infectious disease transmission exist. Inspired by Neural Controlled Differential Equations (NCDE) (Kidger et al., 2020), we exploit a continuous path $\mathbf{Q}_v$ for each region $v$, which reformulates Equation (1) as:

$$\mathbf{Z}(T) = \mathbf{Z}(0) + \int_0^T \psi_t\big(\theta_t; \mathbf{Z}(t)\big)\frac{d\mathbf{Q}(t)}{dt}dt. \tag{2}$$

Equation (2) transforms the integral problem from a Riemann integral to a Riemann-Stieltjes integral. $\mathbf{Q}(t)$ is created from $\{(t_i, \mathbf{x}_i)\}_{i=0}^N$ by an interpolation algorithm.

## 3. Preliminaries

We approach the epidemic forecasting challenge by employing a graph-based prediction model. Let $\mathcal{G} = (\mathcal{V}, \mathcal{E})$ represent the graph, where $\mathcal{V}$ denotes a set of nodes comprising $|\mathcal{V}| = N$ regions (*e.g.*, cities or states). The edge set $\mathcal{E} \subseteq \mathcal{V} \times \mathcal{V}$ represents the geographic links between these regions. The adjacency matrix $\mathbf{A} \in \mathbb{R}^{n \times n}$ is defined such that $\mathbf{A}_{ij} = 1$ if there is an edge $e_{i,j} \in \mathcal{E}$ and $\mathbf{A}_{ij} = 0$ otherwise. The normalized adjacency matrix is given by $\hat{\mathbf{A}} = \mathbf{D}^{-1/2}\mathbf{A}\mathbf{D}^{-1/2}$, where the degree matrix $\mathbf{D}$ is a diagonal matrix with $\mathbf{D}_{ii} = \sum_j \mathbf{A}_{ij}$.

**Problem Formulation.** Each node corresponds to a region with an associated time series input over a window $T$, such as infection counts for $T$ weeks. We represent the training data over this period as $\mathbf{X} = [\mathbf{x}_1, \ldots, \mathbf{x}_T] \in \mathbb{R}^{N \times T}$. The goal is to construct a model capable of predicting an epidemiological profile $\mathbf{x}_{T+h}$ at a future time point $T + h$, where $h$ denotes the prediction horizon.

## 4. Methodology

### 4.1. Overview

In Epidemic-Aware Neural ODE, we initialize disease states as region-specific features and build them upon time variables, which are then fed into proposed Network SIR-inspired neural ODE functions to capture local subtle disease transmission patterns. Furthermore, in Global-guided

Local Transmission Graph we leverage the GNN to obtain global trends and evolutionary graphs. Then dynamic graphs are learned during the evolution of epidemics to guide EANO propagation. Ultimately, we proposed a cross-attention mechanism to accommodate both local nuances and global coherence for final forecasting. The illustration of the overall framework is detailed in Figure 2.

## 4.2. Epidemic-Aware Neural ODE

**Motivation.** Existing deep learning methods neglect the continuous evolution of epidemic systems and do not explicitly learn about epidemic development. Traditional mechanistic models attempt to understand spreading patterns through ODEs but fail to utilize available data sources and model more complex epidemics fully. Therefore, we aim to combine the advantages of both approaches to enable an Epidemic-Aware Neural ODE framework.

**SIR-inspired Neural ODE.** In epidemiology, the standard SIR model (Hethcote, 2000) categorizes the population into three distinct groups based on their disease states: susceptible (S) to infection population, currently infectious population (I), and recovered population (R), with the latter group being immune to both contraction and transmission of the disease. The SIR model, formulated using ODEs (Grassly & Fraser, 2008), describes the epidemic dynamics as follows:

$$\frac{dS}{dt} = -\beta \frac{S_t I_t}{P},$$
$$\frac{dI}{dt} = \beta \frac{S_t I_t}{P} - \gamma I_t, \quad \frac{dR}{dt} = \gamma I_t. \tag{3}$$

These equations distribute the total population $P$ across the aforementioned categories. Here, susceptible individuals become infectious upon contact with infectious ones, driven by the transmission rate $\beta$ ($S \rightarrow I$). In the meanwhile, infectious individuals recover and gain immunity at the recovery rate $\gamma$ ($I \rightarrow R$). However, due to resource restrictions and observation limitations, these explicit cases may not always be available in real-world scenarios. To address this issue, we propose to utilize neural ODEs to automatically infer these ODE functions in Equation (3) via neural networks in a data-driven manner. Specifically, we treat these disease states ($S$, $I$, and $R$) as latent high-dimensional variables. Each state is represented by a matrix $\mathbf{S}(t), \mathbf{I}(t)$, and $\mathbf{R}(t)$ in $\mathbb{R}^{N \times d}$ with $d$ denoting the hidden dimension. In a continuous epidemic system, these states are intrinsically linked with the time variables. Thus, we utilize the NCDE approach in Equation (2) and model the specific epidemic state $\mathbf{C}$ as:

$$\mathbf{C}(T) = \mathbf{C}(0) + \int_0^T \phi_c\Big(\theta_c; \mathbf{C}(t)\Big)\frac{d\mathbf{Z}(t)}{dt}dt,$$
$$= \mathbf{C}(0) + \int_0^T \phi_c\Big(\theta_c; \mathbf{C}(t)\Big)\psi_t\Big(\theta_t; \mathbf{Z}(t)\Big)\frac{d\mathbf{Q}(t)}{dt}dt. \tag{4}$$

Here $\mathbf{Q}(t)$ are controlling paths for regions given by the

interpolation algorithm. They are resilient against irregular cases (*e.g.*, unpredictable outbreaks) when implemented in real-life epidemics, providing a more responsive model for predicting disease spread. Through the NCDE, we can model these disease states in continuous-time epidemics.

Nevertheless, it directly learns these states independently while considering the high-level spatial spreading pattern within diseases. Therefore, we move beyond and designate the ODE function for each state. With well-crafted procedures $\phi_s$, $\phi_i$, and $\phi_r$ functions, we can learn not only intra-state development but also inter-state interactions within regions. Taking SIR process Equation (3) into consideration, $\phi_s$ should be associated as input $\mathbf{S}(t)$ and $\mathbf{I}(t)$, given that $\phi_s\Big(\theta_s; \mathbf{S}(t), \mathbf{I}(t)\Big)$. Similarly, we then have other two functions $\phi_i\Big(\theta_i; \mathbf{S}(t), \mathbf{I}(t)\Big)$ and $\phi_r\Big(\theta_r; \mathbf{I}(t)\Big)$. Inspired by Network SIR applications (Balcan et al., 2009; Sha et al., 2021), we further incorporate regional correlations in disease transmission into the state's updating process. Specifically, these functions are rewritten by:

$$\phi_s = \frac{d\mathbf{S}_v(t)}{dt} = -\mathbf{W}_{\text{trans}}\Big[\mathbf{S}_v(t)|| \sum_{u \in \mathcal{N}_v} \mathbf{e}_{vu}\mathbf{I}_u(t)\Big],$$
$$\phi_i = \frac{d\mathbf{I}_v(t)}{dt} = \mathbf{W}_{\text{trans}}\Big[\mathbf{S}_v(t)|| \sum_{u \in \mathcal{N}_v} \mathbf{e}_{vu}\mathbf{I}_u(t)\Big] - \mathbf{W}_{\text{recov}}\mathbf{I}_v(t),$$
$$\phi_r = \frac{d\mathbf{R}_v(t)}{dt} = \mathbf{W}_{\text{recov}}\mathbf{I}_v(t), \tag{5}$$

here $\mathcal{N}_v$ denotes the neighboring nodes for node $v$ in the set $\mathcal{V}$, with $\mathbf{e}_{vu}$ representing the weight of disease transmission intensity between regions $v$ and $u$. The symbol $||$ signifies the concatenation operation. By substituting the traditional SIR's two simple rates from Equation (3) with the more flexible parameters $\mathbf{W}_{\text{trans}} \in \mathbb{R}^{2d \times d}$ and $\mathbf{W}_{\text{recov}} \in \mathbb{R}^{d \times d}$, our model can derive more detailed representations of the disease spread and recovery processes. This adaptability enables the model to adjust to diverse epidemic conditions, reflecting the intricate mechanisms of disease transmission.

Additionally, incorporating regional correlations through $\mathbf{e}_{vu}$ allows the model to account for spatial dependencies, thereby improving its ability to mirror real-world interactions. However, the approach still depends on static neighborhood relationships and fail to capture the dynamic nature of disease transmission as epidemics evolve.

## 4.3. Global-guided Local Transmission Graph

**Motivation.** As previously noted, EANO solely accounts for the pre-defined neighborhood connections, neglecting the evolutionary disease interactions. This observation drives us to explore more effective approaches to represent local spatial transmission patterns during the evolution.

**Global Infection Trend.** In addition to local transmission

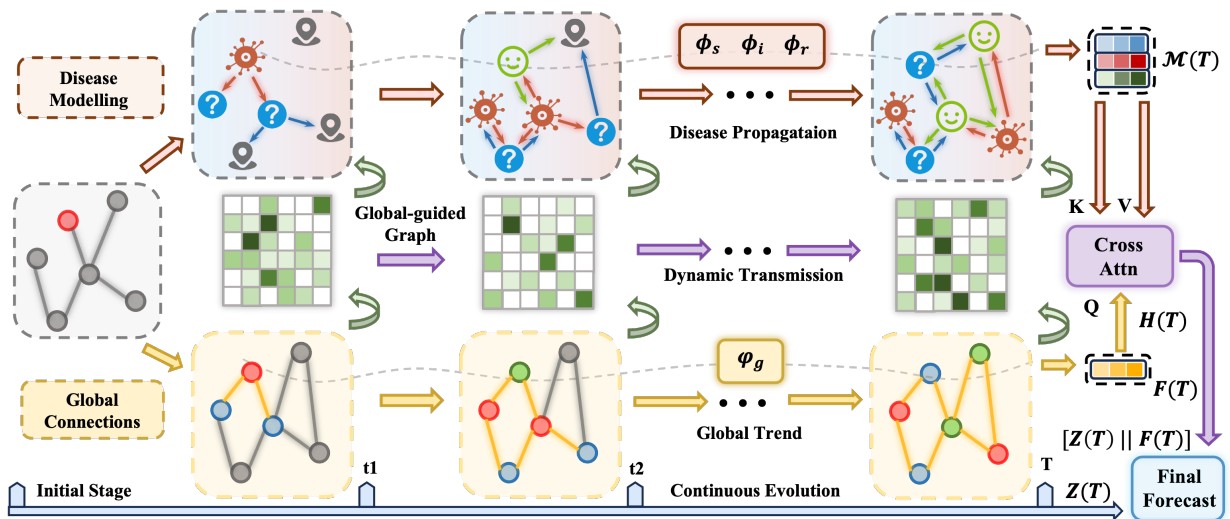

*Figure 2.* Architecture illustration of Epidemiology-Aware Neural ODE with Continuous Disease Transmission Graph. `EARTH` is a general and end-to-end framework that can flexibly capture the time-continuous epidemic mechanism. Best viewed in color.

dynamics, epidemic systems also exhibit a global signal that represents the overall infection trend. This trend encapsulates the broader patterns of infection spread, influenced by factors such as international travel, global health policies, and widespread behavioral changes. For example, during a pandemic, international travel restrictions and lockdowns can significantly curtail cross-border disease transmission, resulting in differing regional infection rates and modifying the epidemic's overall trajectory (Demey et al., 2020; Russell et al., 2021). To address this, we propose creating a global infection indicator feature for each region, correlated with the corresponding temporal features:

$$\mathbf{H}(T) = \mathbf{H}(0) + \int_0^T \varphi_g\Big(\theta_g; \mathbf{H}(t)\Big) \psi_t\Big(\theta_t; \mathbf{Z}(t)\Big) \frac{d\mathbf{Q}(t)}{dt} dt.$$
(6)

Since global here refers to the multi-regional overall direction, we employ Dynamic Time Warping (Please see details in Appendix C) to assess the similarity of historical case trends across different regions, thereby extending the original geographic connections:

$$\tilde{\mathbf{A}}_{uv} = \begin{cases} 1, & \text{if } \mathbf{A}_{uv} = 1, \\ 1, & \text{if } u \neq v \text{ and } u \in Top_k(v), \\ 0, & \text{otherwise.} \end{cases}$$
(7)

In this context, $Top_k(v)$ denotes the indices of the $k$ most similar nodes to node $v$. By doing this, we establish an epidemic-semantic spatial relationship that emphasizes regions with analogous epidemic progression patterns. To promote interactions between regions, we implement a residual GNN layer to update the global infection trend:

$$\varphi_g = \sigma\Big(\tilde{\mathbf{D}}^{-1/2} \tilde{\mathbf{A}} \tilde{\mathbf{D}}^{-1/2} \mathbf{H}(t) \mathbf{W}_g\Big) + \mathbf{H}(t)$$
(8)

Here, $\tilde{\mathbf{D}}$ represents the degree matrix corresponding to $\tilde{\mathbf{A}}$, $\mathbf{W}_g$ is the learnable weight matrix, and $\sigma$ denotes the activation function. The GNN layer $\varphi_g$ facilitates the aggregation

and propagation of information across a broader regional scope, resulting in the fused global trend features $\mathbf{H}(t)$.

**Dynamic Regional Transmission.** Having established the global infection trends, we leverage this signal to guide local spatial transmission patterns by impacting the regional contexts in which local transmission occurs. For instance, high global vaccination rates can decrease the pool of susceptible individuals across regions, reducing local transmission opportunities. We employ a module to learn these dynamically evolving patterns based on $\mathbf{H}(t)$:

$$\mathbf{M}_1(t) = \tanh\Big(\mathbf{H}(t)\mathbf{W}_1 + \mathbf{b}_1\Big),$$

$$\mathbf{M}_2(t) = \tanh\Big(\mathbf{H}(t)\mathbf{W}_2 + \mathbf{b}_2\Big),$$
(9)

$$\tilde{\mathcal{A}}(t) = \sigma\Big(\tanh(\mathbf{M}_1(t)\mathbf{M}_2(t)^\top - \mathbf{M}_2(t)\mathbf{M}_1(t)^\top)\Big).$$

The concurrent local transmission relationship $\tilde{\mathcal{A}}(t)$ is guided by the global trend, while Equation (9) ensures that the learned pattern does not form a completely bidirectional graph. This highlights that inter-regional dissemination is not perfectly symmetrical, capturing the asymmetric nature of spatial interactions in epidemic spread. Additionally, we utilize a masking technique to balance the weights of static and dynamic transmission patterns:

$$\mathbb{M}(t) = \sigma\Big(\mathbf{W}_3 \tilde{\mathcal{A}}(t) + \mathbf{b}_3 \mathbf{J}\Big),$$

$$\mathbf{E}(t) = \mathbb{M}(t) \odot \mathbf{A} + \Big(\mathbf{J} - \mathbb{M}(t)\Big) \odot \tilde{\mathcal{A}}(t).$$
(10)

In Equation (10), $\mathbb{M}(t)$ is a continuous mask matrix, and $\mathbf{J} = \mathbf{1}_N \mathbf{1}_N^\top$ represents the all-ones matrix. The matrix $\mathbf{E}(t)$ integrates the mask matrix $\mathbb{M}(t)$, the original adjacency matrix $\mathbf{A}$, and the globally guided pattern $\tilde{\mathcal{A}}(t)$ through element-wise Hadamard products, aiming to capture propagation patterns between regions in this dynamic system. Once the fused spatial transmission pattern $\mathbf{E}(t)$ is obtained,

| Methods | Australia-COVID | | | | | | US-Region | | | | | | US-States | | | | | |
|---|---|---|---|---|---|---|---|---|---|---|---|---|---|---|---|---|---|---|
| | $h=5$ | | $h=10$ | | $h=15$ | | $h=5$ | | $h=10$ | | $h=15$ | | $h=5$ | | $h=10$ | | $h=15$ | |
| | $\mathcal{R}$ | $\mathcal{P}$ | $\mathcal{R}$ | $\mathcal{P}$ | $\mathcal{R}$ | $\mathcal{P}$ | $\mathcal{R}$ | $\mathcal{P}$ | $\mathcal{R}$ | $\mathcal{P}$ | $\mathcal{R}$ | $\mathcal{P}$ | $\mathcal{R}$ | $\mathcal{P}$ | $\mathcal{R}$ | $\mathcal{P}$ | $\mathcal{R}$ | $\mathcal{P}$ |
| VAR | 665.3 | 83.47 | 575.2 | 92.41 | 502.9 | 95.39 | 1151 | 572.3 | 1396 | 701.6 | 1418 | 688.3 | 339.2 | 90.38 | 371.3 | 103.2 | 402.4 | 150.7 |
| SARIMA | 233.7 | 38.45 | 428.4 | 95.20 | 445.1 | 93.60 | 1142 | 525.8 | 1428 | 715.3 | 1460 | 675.4 | 335.6 | 92.80 | 368.2 | 106.4 | 406.5 | 149.2 |
| SIR | 235.0 | 42.50 | 445.2 | 104.5 | 425.6 | 102.8 | 1235 | 575.0 | 1480 | 745.0 | 1565 | 760.0 | 338.0 | 121.5 | 388.0 | 112.0 | 428.0 | 168.0 |
| LSTM | 228.0 | 39.78 | 433.6 | 108.4 | 432.6 | 98.09 | 1173 | 538.6 | 1475 | 736.5 | 1509 | 757.6 | 331.8 | 93.45 | 370.0 | 109.8 | 411.3 | 154.9 |
| DCRNN | 514.8 | 166.5 | 853.7 | 286.4 | 1186 | 404.6 | 1488 | 760.9 | 1443 | 732.7 | 1412 | 710.3 | 329.4 | 93.15 | 334.7 | 96.90 | 372.8 | 142.6 |
| STGCN | 833.7 | 232.6 | 787.8 | 227.7 | 802.1 | 248.3 | 1335 | 678.1 | 1522 | 819.2 | 1638 | 925.4 | 304.7 | 89.32 | 293.7 | 85.33 | 312.5 | 116.3 |
| ASTGCN | 821.5 | 221.9 | 765.9 | 201.3 | 804.1 | 254.8 | 1252 | 545.2 | 1478 | 801.1 | 1576 | 821.3 | 310.2 | 93.44 | 290.5 | 80.99 | 344.6 | 123.4 |
| STGODE | 310.5 | 66.32 | 392.2 | 91.05 | 571.3 | 159.2 | 1304 | 668.2 | 1403 | 732.1 | 1577 | 804.3 | 345.2 | 107.8 | 402.4 | 120.4 | 477.3 | 199.4 |
| STG-NCDE | 287.2 | 49.21 | 341.3 | 77.92 | 479.2 | 111.2 | 1284 | 643.1 | 1399 | 691.2 | 1421 | 732.1 | 319.2 | 94.39 | 377.6 | 101.5 | 421.3 | 176.7 |
| CNNRNN-Res | 1802 | 624.7 | 612.6 | 151.4 | 622.1 | 153.1 | 1190 | 588.3 | 1332 | 642.8 | 1374 | 652.1 | 303.3 | 86.78 | 292.1 | 79.33 | 333.6 | 105.4 |
| EpiGNN | 210.3 | 40.12 | 467.3 | 120.1 | 764.2 | 233.7 | 1136 | 534.2 | 1454 | 728.9 | 1444 | 764.2 | 288.5 | 84.32 | 297.6 | 84.32 | 391.6 | 157.4 |
| CAMul | 231.4 | 44.32 | 398.2 | 76.62 | 634.1 | 164.7 | 1145 | 557.3 | 1434 | 703.2 | 1402 | 699.2 | 294.6 | 88.16 | 312.8 | 86.71 | 325.2 | 107.5 |
| EINN | 206.2 | 38.19 | 312.4 | 64.21 | 456.9 | 98.72 | 1178 | 571.6 | 1432 | 729.1 | 1489 | 792.3 | 321.2 | 97.91 | 342.1 | 100.1 | 402.7 | 162.9 |
| ColaGNN | 224.2 | 55.23 | 544.8 | 161.6 | 795.8 | 258.0 | 1148 | 533.6 | 1524 | 846.6 | 1552 | 856.3 | 299.1 | 81.53 | 283.4 | 79.12 | 339.4 | 120.6 |
| EpiColaGNN | 204.3 | 36.86 | 345.4 | 68.39 | 886.0 | 296.5 | 1185 | 575.7 | 1341 | 648.1 | 1371 | 666.9 | 286.1 | 83.38 | 300.9 | 90.65 | 375.1 | 132.5 |
| EARTH | **156.8** | **30.12** | **177.6** | **38.62** | **225.3** | **56.32** | **1080** | **522.4** | **1244** | **605.3** | **1301** | 647.1 | **243.2** | 67.43 | **277.8** | 80.43 | **300.1** | **104.2** |

*Table 1.* **Comparison with the state-of-the-art methods** on three epidemic forecasting datasets. Best in bold and second with underline.

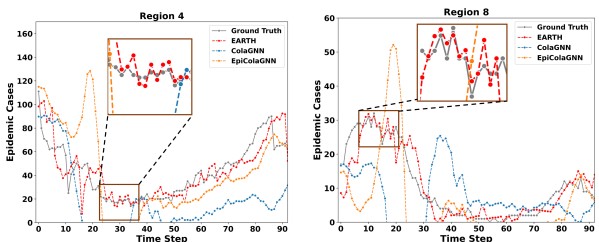

(a) Visualization of Region 4   (b) Visualization of Region 8

*Figure 3.* Visualization of predicted cases. We randomly pick two regions in the Australia-COVID dataset with horizon 10. It shows that EARTH fits the ground truth well and follows the developing trend of epidemics. Better view in enlarged.

we use it to update the weights $\mathbf{e}_{uv}$ in Equation (5), resulting in:

$$\phi_s = -\mathbf{W}_{\text{trans}}\Big[\mathbf{S}_v(t)||\sum_{u\in\tilde{\mathcal{N}}_v}\mathbf{e}_{vu}(t)\mathbf{I}_u(t)\Big],$$

$$\phi_i = \mathbf{W}_{\text{trans}}\Big[\mathbf{S}_v(t)||\sum_{u\in\tilde{\mathcal{N}}_v}\mathbf{e}_{vu}(t)\mathbf{I}_u(t)\Big] - \mathbf{W}_{\text{recov}}\mathbf{I}_v(t), \quad (11)$$

$$\phi_r = \mathbf{W}_{\text{recov}}\mathbf{I}_v(t).$$

The continuous and regional correlation intensity is denoted by $\mathbf{e}_{uv}(t)$. The set $\tilde{\mathcal{N}}_v$ represents the neighborhood nodes with nonzero weights in $\tilde{\mathcal{A}}(t)$ for the dynamic regional transmission of region $v$. This approach enables GLTG to leverage the global infection trend signal to guide local spatial transmission patterns, considering diverse external factors in epidemics and extending beyond simple partial transfer.

**Global and Local Epidemic Fusion.** With the differential and integral processes of epidemics established, we determine the global infection trend $\mathbf{H}(t)$ for continuous time $t$

using a designated ODE solver, such as *Runge–Kutta*:

$$\mathbf{H}(t) = \text{ODESolver}\left(\frac{d\mathbf{H}(t)}{dt}, \mathbf{H}_0, t\right). \quad (12)$$

Given the overall historical time window $T$, we ultimately derive the global infection trend $\mathbf{H}(T)$ and local disease states $\mathcal{M}(T) = \{\mathbf{S}(t), \mathbf{I}(t), \mathbf{R}(t)\}$. To integrate both the global coherence of epidemic trends and the local intricacies of disease states, we design a multi-headed cross-attention mechanism to merge the global and local transmission information. Specifically, we use $\mathbf{H}(T)$ to guide the fusion of $\mathcal{M}(T)$. Given three common sets of inputs: query set $Q$, key set $K$, and value set $V$, we define $\mathcal{H}$ as follows:

$$\mathcal{H}(Q, K, V) = (\Omega_1 \oplus \Omega_2 \oplus \cdots \oplus \Omega_{N_\mathcal{T}})\mathbf{W},$$

$$\mathcal{S}(Q, K, V) = \text{softmax}\left(\frac{QK^T}{\sqrt{d_f}}\right)V, \quad (13)$$

$$\Omega_\mu = \mathcal{S}(Q\mathbf{W}\mu^Q, K\mathbf{W}\mu^K, V\mathbf{W}\mu^V)|_{\mu=1}^{N_\mathcal{T}}.$$

The $\mu$-th head is represented by $\Omega_\mu$, and the attention function is denoted as $\mathcal{S}$. The learnable linear mappings include $\mathbf{W}, \mathbf{W}^Q, \mathbf{W}^K,$ and $\mathbf{W}^V$. The formulation for the global-local fusion is given by:

$$\mathbf{F}(T) = \mathcal{H}\Big(\mathbf{Z}(T), \mathcal{M}(T), \mathcal{M}(T)\Big). \quad (14)$$

In Equation (14), $\mathbf{F}(T)$ represents the fused feature. Conceptually, the global trend feature serves as a query, calculating the similarity with each detailed disease state. This method aids in recognizing the semantic epidemic conditions and attentively integrating the disease features.

| Missing Rate | Australia-COVID | | | | US-Region | | | |
| --- | --- | --- | --- | --- | --- | --- | --- | --- |
| | $h = 5$ | | $h = 10$ | | $h = 5$ | | $h = 10$ | |
| | $\mathcal{R}$ | $\mathcal{P}$ | $\mathcal{R}$ | $\mathcal{P}$ | $\mathcal{R}$ | $\mathcal{P}$ | $\mathcal{R}$ | $\mathcal{P}$ |
| 40% | 173.1 | 38.02 | 190.2 | 45.68 | 1129 | 541.9 | 1267 | 629.9 |
| 30% | 168.5 | 36.42 | 187.2 | 44.50 | 1115 | 535.4 | 1265 | 631.8 |
| 20% | 162.4 | 33.44 | 184.7 | 42.97 | 1110 | 536.2 | 1262 | 618.4 |
| 10% | 158.4 | 30.95 | 180.4 | 40.77 | 1089 | 529.6 | 1259 | 613.1 |
| 0% | 156.8 | 30.12 | 177.6 | 38.62 | 1080 | 522.4 | 1244 | 605.3 |

*Table 2.* **Analysis under irregular conditions** on two datasets.

### 4.4. Overall Objective

Ultimately, we derive the fused features $\mathbf{F}(T)$, which harmonize the global consistency of epidemic trends with the particularities of local health conditions. We then concatenate these features with the time-corresponding features $\mathbf{Z}(T)$ and employ an MLP parameterized by $\theta_f$ to pool the final prediction for region $v$:

$$y_v = f(\theta_f; [\mathbf{F}_v(T) || \mathbf{Z}_v(T)]). \tag{15}$$

Following previous methods (Deng et al., 2020; Xie et al., 2022), we use the MSE loss to compare the predicted values with the ground truth:

$$\mathcal{L}_{mse} = \sum_{i=1}^{B} \sum_{v=1}^{N} |y_{i,v} - \hat{y}_{i,v}|, \tag{16}$$

where $B$ denotes the sample size, and $i$ is the sample index. $\hat{y}_{i,v}$ represents the true value for sample $i$ of region $v$.

## 5. Experiment

In this section, we comprehensively evaluate our proposed EARTH by answering the main questions:

- **Q1: Performance.** Does EARTH outperforms the existing state-of-the-art epidemic forecasting methods?
- **Q2: Resilience.** Is EARTH stable on different settings?
- **Q3: Effectiveness.** Are proposed two key components: EANO and GLTG both effective?
- **Q4: Sensitivity.** What is the performance of the proposed method with different hyper-parameters?

The answers of **Q1-Q4** are illustrated as follows.

### 5.1. Experimental Setup

**Real-world Datasets.** We leverage three datasets to examine the validity of our EARTH, to ensure fair and consistent comparisons with prior work (Liu et al., 2023; Kamarthi & Prakash), including COVID-19 and influenza-like illness: Australia-COVID, US-Regions, and US-States. Please see Appendix A for dataset details.

**Implementation Details.** We use two metrics following (Liu et al., 2023): $\mathcal{R}$ represents RMSE (Root Mean Square Error), while $\mathcal{P}$ stands for Peak Time Error, which calculates the MAE (Mean Absolute Error) focusing only on significant peaks in the epidemics using a specified threshold. For more details please refer to Appendix B.

**Counterparts.** We compare ours against several SOTA epidemic forecasting methods: VAR (Song et al., 2020), SARIMA (Valipour, 2015), SIR (Grassly & Fraser, 2008), LSTM (Sesti et al., 2021), DCRNN (Li et al., 2018), STGCN (Yu et al., 2017), ASTGCN (Guo et al., 2019), STGODE (Fang et al., 2021), STG-NCDE (Choi et al., 2021), CNNRNN-Res (Wu et al., 2018), CAMul (Kamarthi et al., 2021), EINN (Rodríguez et al., 2023), Co-laGNN (Deng et al., 2020), EpiGNN (Xie et al., 2022) and EpiColaGNN (Liu et al., 2023).

### 5.2. Performance

This section addresses **Q1**. To demonstrate the excellent performance of our proposed EARTH, we conducted comprehensive experiments on various epidemic datasets. We considered multiple baselines, including general spatio-temporal and epidemic forecasting methods, as detailed in Tab. 1. Key observations include: 1) VAR and LSTM are inadequate at capturing complex spatial dependencies, making them less effective. 2) General spatio-temporal methods like STGCN or ASTGCN can capture some regional dependencies but struggle with time development and long-term predictions. 3) ODE-based methods like STGODE can learn complex dynamic systems but do not sufficiently consider epidemic mechanisms. 4) Some epidemic-specific methods achieve excellent results but still struggle to model the evolution of epidemics. 5) Mechanistic methods like EINN do not succeed in capturing high-level spatial interaction between diseases from different regions. 6) EARTH demonstrates competitive performance across various real-world datasets due to its ability to learn complex epidemic evolution and dynamic regional propagation patterns.

Additionally, to visually underscore the superiority of EARTH, we compared predicted cases in the Australia-COVID dataset with a horizon of 10 against different baselines. The results, shown in Figure 3, indicate that our method more accurately fits the ground truth and follows the trend of epidemic development. We also show the learned graph in our GLTG component in Figure 4, which demonstrates that our method goes beyond geographical connections and obtains global horizons during evolution.

### 5.3. Resilience

This section addresses the question **Q2**. We conducted two key experiments to evaluate this aspect: 1) We examine the robustness of the method under different irregular conditions with a range of missing rates, as detailed in Tab. 2. The

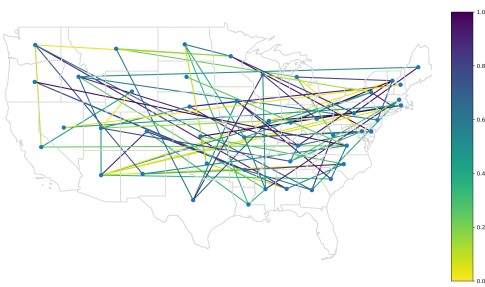

*Figure 4.* **Learned Regional Graph** in GLTG. We visualize top-3 weighted edges for each region in the US-States dataset, excluding states with no available data.

outcomes show that `EARTH` remains robust across different datasets with diverse intervals. Our proposed method consistently outperforms other baseline methods, demonstrating its resilience to variable intervals and missing data. 2) We also test the performance of `EARTH` across different prediction horizons as shown in Figure 6. The results indicate that our method can make stable predictions over various horizons while learning epidemic mechanisms enables superior long-term prediction compared to other methods.

### 5.4. Effectiveness

The explanation for **Q3** is presented in this section. Tab. 3 firstly discusses two key design elements in our method: EANO effectively enhances performance by leveraging the powerful capabilities of Neural ODEs while specifically considering the disease propagation mechanism. Additionally, GLTG yields promising results by learning the dynamic regional patterns during the evolution of epidemics.

In addition, we dive into GLTG deeper by considering it without dynamic graphs (w/o Dyna. Graph) or global trends (w/o Glo. Trend). The results show that dynamic graphs play a crucial role in modeling disease spatial interactions, while global trends help to capture long-distance information. Additionally, We provide two variants: one using concatenation (w Concat) and another using addition (w Add) instead of cross-attention. These variants show the effectiveness and necessity of cross-attention, as it outperforms both by selectively emphasizing relevant global and local information. We also examine `EARTH` under a fully connected regional graph, declaring this will lead to information redundancy. The variant considering adding a sparse penalty loss to the learned graph for encouraging sparsity, will not impact the final performance significantly.

### 5.5. Sensitivity

This section provides an answer to **Q4**. As shown in Figure 5, we first investigate the effect of varying the number of attention heads $N_{\mathcal{T}}$ in Equation (13). The results demonstrate overall stability with different numbers of heads, al-

|  | Australia-COVID | | | | US-Region | | | |
|---|---|---|---|---|---|---|---|---|
| Variants | $h=5$ | | $h=10$ | | $h=5$ | | $h=10$ | |
|  | $\mathcal{R}$ | $\mathcal{P}$ | $\mathcal{R}$ | $\mathcal{P}$ | $\mathcal{R}$ | $\mathcal{P}$ | $\mathcal{R}$ | $\mathcal{P}$ |
| w/o Both | 267.4 | 43.65 | 322.7 | 63.04 | 1235 | 637.4 | 1367 | 675.3 |
| w EANO | 178.6 | 36.99 | 184.5 | 44.62 | 1120 | 538.3 | 1282 | 639.2 |
| w GLTG | 232.4 | 40.44 | 301.2 | 57.42 | 1204 | 579.3 | 1321 | 654.3 |
| w/o Dyna. Graph | 227.6 | 38.44 | 290.0 | 53.89 | 1184 | 572.5 | 1302 | 647.0 |
| w/o Glo. Trend | 172.4 | 33.75 | 182.0 | 44.39 | 1102 | 541.2 | 1267 | 621.3 |
| w Concat | 167.9 | 33.44 | 190.4 | 45.71 | 1132 | 544.3 | 1304 | 612.4 |
| w Add | 175.2 | 33.01 | 194.8 | 43.29 | 1149 | 535.7 | 1291 | 603.7 |
| Fully Connected | 192.1 | 39.62 | 194.7 | 40.22 | 1192 | 564.3 | 1347 | 666.0 |
| Sparse Penalty | 160.9 | 32.60 | 182.4 | 59.37 | 1075 | 519.2 | 1271 | 635.4 height |
| `EARTH` | 156.8 | 30.12 | 177.6 | 38.62 | 1080 | 522.4 | 1244 | 605.3 |

*Table 3.* **Ablation Study of different variants** on two datasets.

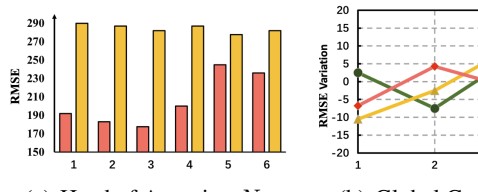

(a) Head of Attention $N_{\mathcal{T}}$     (b) Global Connection $k$

*Figure 5.* **Analysis on hyper-parameter.** Performance with hyper-parameter $N_{\mathcal{T}}$ and $k$, where red, yellow, and green represent the Australia-COVID, US-States, and US-Region respectively.

though too few heads can impair the method's ability to capture diverse information. We also test our method with different values of $k$ in Equation (7). More connections can lead to redundancy in message passing for datasets with fewer regions (*e.g.*, Australia-COVID). In contrast, larger datasets (*e.g.*, US-States) can tolerate more connections. In all cases, global connections are essential for learning global infection trends.

## 6. Conclusion

In this paper, we propose a novel framework, `EARTH`, to improve epidemic forecasting performance. By integrating neural ODEs with traditional compartmental models, EANO captures the underlying disease propagation mechanisms. We also identify global infection trends and introduce GLTG to dynamically adjust local transmission patterns. Using a global-local cross-attention fusion approach, we extract representative features that account for both subtle disease states and broader trends. Extensive experiments on real-world epidemic datasets highlight the effectiveness of `EARTH`, offering valuable insights into combining mechanistic models with deep learning for future applications in epidemiology and data science.

## Impact Statement

This paper presents work whose goal is to advance the field of Machine Learning. There are many potential societal consequences of our work, none of which we feel must be

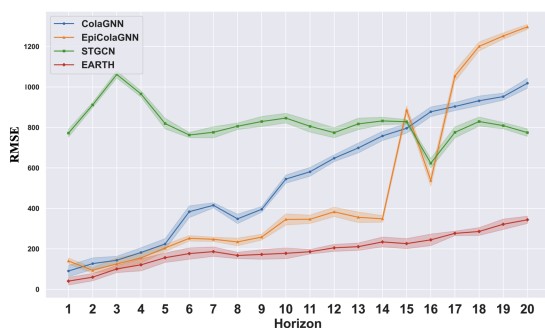

*Figure 6.* **Analysis on different horizon** with four methods.

specifically highlighted here.

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

## A. Datasets Details

Following previous epidemic forecasting work (Deng et al., 2020; Liu et al., 2023), we exploit three widely-used datasets including COVID-19 and influenza-like illness:

- **Australia-COVID.** Provided by JHU-CSSE[1], this dataset records daily new COVID-19 cases, including 6 states and 2 territories, from January 2020 to August 2021.
- **US-Regions.** This dataset comprises weekly influenza activity levels for 10 Health and Human Services (HHS) regions, spanning from 2002 to 2017, and offers data on regional influenza patterns over time.
- **US-States.** The US-States dataset contains weekly counts of patient visits for influenza-like illness (ILI) across 49 states in the United States from 2010 to 2017, excluding Florida, capturing influenza trends.

## B. Implemention Details

In all experimental setups, we set the learning rate to $1e - 3$ and use *SGD* (Robbins & Monro, 1951) as the optimizer with a momentum of 0.9 and weight decay of $1e - 5$ (Bi et al., 2025a;b). The default hidden size is 64, and the window size $T$ is 20. Considering that decision-makers need time to allocate prevention resources in epidemic modeling, we set the horizon $h$ to 5, 10, and 15. We repeat each experiment five times for each dataset and record the average results.

## C. Details about Dynamic Time Warping

In our method, we use Dynamic Time Warping (DTW) to measure similarity between nodes based on historical case trends. The DTW distance between two time series $X = \{x_1, x_2, \ldots, x_n\}$ and $Y = \{y_1, y_2, \ldots, y_m\}$ is computed by constructing an $n \times m$ cumulative distance matrix $D$, where each element $D(i, j)$ represents the minimum cumulative distance between the first $i$ elements of $X$ and the first $j$ elements of $Y$, with $d(x_i, y_j) = \|x_i - y_j\|$ as the Euclidean distance. The recursive relation is:

$$D(i, j) = d(x_i, y_j) + \min\{D(i - 1, j), D(i, j - 1), D(i - 1, j - 1)\}. \tag{17}$$

Finally, $D(n, m)$ gives the DTW distance between $X$ and $Y$. Using this DTW metric, we define the $k$-most similar nodes to node $v$ as:

$$\text{Top}_k(v) = \{u \mid u \in V, D(X_u, X_v) \text{ is among the } k \text{ smallest values}\}. \tag{18}$$

This approach captures alignment in epidemic patterns across regions, even with temporal shifts. By identifying the $k$-most similar nodes using DTW, we create a flexible, semantically meaningful connection structure that enhances model performance.

---

[1]https://github.com/CSSEGISandData/COVID-19

