# OpenReview forum: "EARTH: Epidemiology-Aware Neural ODE with Continuous Disease Transmission Graph"
_ICML.cc/2025/Conference — ICML 2025 poster_

### Official Review · Reviewer_XKTr · 2025-03-11

**Overall Recommendation:** 5

**Summary:**

This paper introduces a new epidemic forecasting framework, EARTH, which combines neural ODEs with traditional compartmental models. The core idea behind EARTH is to address common forecasting challenges such as irregular data sampling and missing values by integrating two main modules: a local transmission model based on Epidemic-Aware Neural ODEs and a Global-guided Local Transmission Graph that incorporates global trends. Through experiments on real-world datasets, like COVID-19 data, the authors demonstrate that EARTH achieves superior performance compared to existing methods, suggesting that it could play a crucial role in enhancing the accuracy of epidemic prediction.

## update after rebuttal

The authors' responses have addressed my concerns. After reviewing the comments as well as the discussions, I'd like to raise my score to 5.

**Claims And Evidence:**

The paper’s claims are supported by a variety of evidence that extends beyond just experimental results. 1) Experimentally, the model is evaluated on several real-world datasets—including COVID-19 data—with consistent improvements over baselines in metrics such as RMSE and MAE. 2) Detailed ablation studies demonstrate that both the neural ODE component and the global transmission graph are crucial to achieving high performance, as their removal significantly degrades the results. 3) Additionally, hyperparameter sensitivity tests confirm that the model performs robustly across a range of settings, and clear visualizations of predicted versus actual trends further validate its practical applicability.

**Essential References Not Discussed:**

The paper already does a good job of addressing relevant literature. It connects well with existing research in epidemic modeling and machine learning.

**Experimental Designs Or Analyses:**

The experimental designs and analyses in this paper are robust and well thought out. The authors employ multiple real-world datasets, including COVID-19 data, and standard metrics such as RMSE and MAE to evaluate performance. Comprehensive ablation studies in Table 3 effectively isolate the contributions of each model component, while sensitivity analyses  from Figure 5 confirm the stability of the model across various hyperparameters. The clear visualizations in Figure 3 and Figure 4 further support the validity of the results by illustrating how the model captures key trends.

**Methods And Evaluation Criteria:**

The proposed methods and evaluation criteria are well-suited to the problem. Additionally, using real-world datasets like COVID-19 data with standard metrics such as RMSE and MAE offers a robust framework for evaluation.

**Other Comments Or Suggestions:**

Please ref to weaknesses.

**Other Strengths And Weaknesses:**

Strengths:

1)	The paper introduces an innovative approach by combining neural ODEs with traditional epidemic models and incorporating a Global-guided Local Transmission Graph, which effectively captures both continuous dynamics and spatial transmission patterns.

2)	The authors provide clear motivation for integrating machine learning with classical epidemiological frameworks, addressing limitations of traditional models in real-world epidemic forecasting.

3)	This work advances the field by demonstrating how modern deep learning techniques can be seamlessly merged with established epidemic models, paving the way for more accurate and dynamic forecasting tools.

4)	By bridging the gap between classical epidemiological modeling and contemporary machine learning, the paper makes a significant contribution to computational epidemiology and opens up new research avenues for future advancements in the field.

Weaknesses:

1)	There is no detailed analysis of the computational complexity, which could help assess the method's scalability and efficiency in large-scale applications.

2)	The model may be sensitive to certain hyperparameters (such as learning rate, hidden layer dimensions, regularization coefficients, etc.). Although the author conducted experiments, different hyperparameter settings may affect the model's generalization ability and stability. Further experiments can explore the impact of hyperparameter adjustment on model performance.

**Questions For Authors:**

-	How are these missing rates introduced into the data? Are they intentionally sampled deletions from the original data (such as random deletions) to simulate missingness, or do they originate from missing data in real-world scenarios?
-	In Figure 4, are there any isolated nodes with almost no connections to other nodes? If there are, what does this mean?

**Relation To Broader Scientific Literature:**

In terms of machine learning, the paper draws on work related to neural differential equations and graph neural networks. While these techniques have been applied successfully in other domains, their application to epidemic forecasting is a significant innovation. By leveraging these advanced techniques, the paper contributes to the growing body of research on using deep learning to enhance predictive modeling in epidemiology. This approach also connects to broader trends in the literature about the fusion of classical and modern machine learning techniques to address real-world problems.

**Theoretical Claims:**

I reviewed the theoretical claims, especially those related to the integration of neural ODEs with classical epidemic models. The manuscript’s mathematical formulation is generally free from notable errors, and the conceptual foundations are sound and well-motivated. While the discussion primarily leverages established principles without extensive formal proofs, the theoretical framework is sufficiently robust, and the empirical results further reinforce its validity. Additionally, including an analysis of computational complexity could provide deeper insights into the scalability and practical applicability of the proposed methods, but this is a minor point.

---

> ### Author Rebuttal · Authors · 2025-03-31
>
> # Response to Reviewer XKTr
>
> We sincerely thank you for your thorough review and positive assessment of our work. We are grateful for your recognition of EARTH's
> novelty and contributions to epidemic forecasting. We address your questions below:
>
> > `Weakness 1`: No detailed analysis of the computational complexity.
>
> We appreciate this valuable suggestion. The computational complexity of EARTH can be broken down into:
>
> - **Epidemic-Aware Neural ODE (EANO)**: The computational complexity of our neural ODE solver is primarily determined by $O(T_{\text{ODE}} \times (N \times d^2 + |E| \times d))$, where $T_{\text{ODE}}$ represents the average number of solver steps needed for integration, $N$ is the number of regions, $d$ is the hidden dimension, and $|E|$ is the number of transmission edges in our graph. The term $N \times d^2$ comes from node-level feature transformations applied at each solver step, while $|E| \times d$ reflects the message passing operations between connected regions during epidemic propagation.
>
> - **Global-guided Local Transmission Graph (GLTG)**: This component has a theoretical complexity of $O(N^2 \times T_{\text{hist}}^2 + N^2 \times d)$. The first term arises from computing pairwise temporal similarities using Dynamic Time Warping across $N$ regions with historical sequences of length $T_{\text{hist}}$, while the second term corresponds to the generation of the full adjacency matrix with feature transformations. In practice, we reduce this cost using FastDTW for efficient similarity calculation and by maintaining a sparse graph structure that retains only the most relevant region connections.
>
> - **Cross-Attention Mechanism**: The complexity for our cross-attention operation between epidemic states and global features is $O(N \times d^2)$, which is notably more efficient than typical attention mechanisms that scale with $O(N^2 \times d)$. This efficiency stems from our design choice to constrain attention to the three epidemic states (S,I,R) per region rather than attending across all regions.
>
> We will include this detailed analysis in the revised version.
>
> > `Weakness 2`: Model sensitivity to hyperparameters.
>
> Thank you for this important observation. We have conducted additional experiments to evaluate EARTH's sensitivity to key hyperparameters on the Australia-COVID dataset with a horizon of 5:
>
> | Hidden Dimensions | 16 | 32 | 64 | 128 | 256 |
> |-------------------|-----|-----|-----|------|------|
> | RMSE | 187.3 | 176.5 | 156.8 | 159.2 | 172.8 |
> | MAE | 42.64 | 36.27 | 30.12 | 31.95 | 38.76 |
>
> | Learning Rate | 5e-5 | 1e-4 | 5e-4 | 1e-3 | 5e-3 |
> |---------------|------|------|------|------|------|
> | RMSE | 179.1 | 167.5 | 160.3 | 156.8 | 163.2 |
> | MAE | 41.56 | 36.28 | 32.41 | 30.12 | 33.95 |
>
> These results demonstrate that EARTH performs optimally with hidden dimensions of 64 and a learning rate of 1e-3, which aligns with our main experimental setup in Table 1. We will incorporate these detailed sensitivity analyses in the revised manuscript.
>
> > `Question 1`: How are these missing rates introduced into the data?
>
> Thank you for this question about our experimental methodology.
> For the controlled experiments in Table 2, we artificially introduced missingness by randomly removing data points from the complete dataset at rates of 10%-40%. This random deletion strategy follows standard practice in the literature for evaluating model robustness to missing data.
>
> > `Question 2`: In Figure 4, are there any isolated nodes with almost no connections to other nodes? If there are, what does this mean?
>
> This is an insightful question. What appears as isolated nodes in Figure 4 is simply a result of our visualization approach - we only display the top-3 highest weighted edges for each region to maintain visual clarity. In our Local Transmission Graph, we first use Dynamic Time Warping to select top-k similar nodes based on temporal epidemic patterns, then adaptively learn the normalized weighted edges during training using the mechanisms described in Equations 9-10, allowing EARTH to automatically determine appropriate information sharing between regions based on epidemic similarity. We will clarify this visualization choice in the revised manuscript.

---

> > ### Comment · Reviewer_XKTr · 2025-04-02
> >
> > Thanks for your rebuttal. My concerns have been well addressed, and I've checked other reviewers' feedback as well. I suggested that the author include complexity analysis and hyperparameter studies in the revised manuscript. We're at an inflection point for AI in computational epidemiology, and the work may have a potential impact. I keep my score and vote for acceptance.

---

> > > ### Author Response · Authors · 2025-04-02
> > >
> > > # Response to Reviewer XKTr
> > > Dear Reviewer XKTr,
> > >
> > > **Thank you again for recognizing the innovation and contribution of our work and for your willingness to support its acceptance!**
> > >
> > > Best regards,
> > >
> > > Authors

---

### Official Review · Reviewer_rfdg · 2025-03-11

**Overall Recommendation:** 3

**Summary:**

The authors tackle the problem of effectively forecasting epidemics and propose the so-called EARTH method, which combines an epidemiology-aware neural ODE with a continuous disease transmission graph. More specifically, they leverage a **neural ODE-based component** (EANO) based on the **common epidemic SIR mechanism** to capture spatial spreading during disease evolution while enabling continuous modeling. They also introduce a **GNN-based component that captures global epidemic trends** (GLTS) and a cross-attention mechanism that combines global patterns with local transmission patterns. The proposed method is evaluated on real-world datasets capturing COVID-19 and influenza diseases against other baseline methods, showing performance improvements for few horizon lengths under node regression metrics (such as RMSE).

## update after rebuttal

The authors have addressed some of my initial concerns, particularly those related to the significance of results and experimental design details. However, critical issues around the theoretical positioning (against the relevant first works in the field), selection and interpretability of learned epidemic rates, and justification for explicit SIR-based modeling remain only partially resolved and should be clearly reflected in the revised manuscript. Time/memory cost comparisons against baselines are still missing and can be significant. I revise my recommendation to weak accept, assuming the authors will incorporate these clarifications and adjustments into the final version.

**Claims And Evidence:**

The following claims of the paper are problematic:
- Introduction, page 2: *“We are the first to harmonize the neural ODE with the epidemic mechanism [...] patterns”*. The authors claim that they are the first to combine the Neural ODE method with approximate system equations for epidemic spreading. However, it seems that relevant works in this field have already combined ODEs from epidemic compartmental models with advances in neural ODE solvers, such as in [1].
- Methodology, page 4: *“By substituting the traditional SIR’s two simple rates [...] our model can derive more detailed representations of the disease spread and recovery processes.”*
The infection $\beta$ and recovery $\gamma$ rates in SIR models are fundamental parameters in epidemic modeling and have been extensively studied to develop realistic simulations of disease spread. However, the authors do not showcase how learning these parameters improves performance.

1. Kosma, C., Nikolentzos, G., Panagopoulos, G., Steyaert, J. M., & Vazirgiannis, M. (2023). Neural ordinary differential equations for modeling epidemic spreading. Transactions on Machine Learning Research.

**Essential References Not Discussed:**

Several methods in the area of physics-informed modelization of epidemic spreading (particularly based on compartmental models, e.g., SIR, SEIR) could be included in the introduction/related work of the paper to support the positioning of its main concepts. Some examples:
- *SIR Neural ODEs on Networks:* Kosma, C., Nikolentzos, G., Panagopoulos, G., Steyaert, J. M., & Vazirgiannis, M. (2023). Neural ordinary differential equations for modeling epidemic spreading. Transactions on Machine Learning Research.
- *SIR-based Embedding Layers:* Zheng, Y., Jiang, W., Zhou, A., Hung, N. Q. V., Zhan, C., & Chen, T. (2024). Epidemiology-informed Graph Neural Network for Heterogeneity-aware Epidemic Forecasting. arXiv preprint arXiv:2411.17372.
- *Physics-Informed Neural Networks:* Cai, M., Em Karniadakis, G., & Li, C. (2022). Fractional SEIR model and data-driven predictions of COVID-19 dynamics of Omicron variant. Chaos: an interdisciplinary journal of nonlinear science, 32(7).
- *Dynamic Message Passing for SIR combined with GNNs:* Gao, F., Zhang, J., & Zhang, Y. (2022, May). Neural enhanced dynamic message passing. In International Conference on Artificial Intelligence and Statistics (pp. 10471-10482). PMLR.

**Experimental Designs Or Analyses:**

The common experimental design choices with the cited studies in section 5.1 are unclear. More specifically, different horizon lengths/window sizes are used in studies (Liu et al., 2023, Kamarthi & Prakash) and the proposed method, while it seems that a common choice in time series forecasting is to consider multiples of the input window length (e.g., 1W, 2W, 3W,...).

**Methods And Evaluation Criteria:**

This study leverages common baselines and benchmark datasets in the field. Evaluation metrics (point-wise deviation) are also common.

**Other Comments Or Suggestions:**

Not applies.

**Other Strengths And Weaknesses:**

*Strengths:*
- The paper is well-written and easy to follow.
- The experimental results consider several baseline modes and thorough ablation studies of the proposed method's main components.

*Weaknesses:*
- **S1** - *Presentation and Design choices of the epidemic compartment:* In real-world scenarios, epidemic data is often noisy. If $\beta$ and $\gamma$ are learned directly from data without constraints, the model might overfit short-term trends rather than capturing realistic transmission dynamics. Necessary assumptions to derive this form of equations (3), intuition, and limitation of the compartmental model in practical applications are not discussed.
- **S2** - *Positioning against works explicitly combining ML and compartmental modes:* Based on the missing references above, the presentation of related works fails to showcase the limitations of existing approaches and design choices on incorporating compartmental models/spreading priors to the loss function/architectural structure of the methods.
- **S3** - *Experimental Choices not well-justified/missing:*
1. It is unclear why the authors choose h=5,10,15 as horizon lengths for their experiments, given a historical window of 20 timestamps, which constitutes the task rather easy. A study of the performance impact with larger (>20) forecasting horizons would be interesting.
2. The type (e.g., runge-kutta) and selection of step of the solver chosen in the neural-ode are not mentioned.
3. Standard deviations are not mentioned.
4. The downsampling method followed for synthetically creating irregular timestamps in the datasets is unclear but can have a significant impact on the distortion of the underlying continuous dynamics.
- **S4** - *Computational Analysis is missing:* The proposed method relies on computationally heavy components, including the neural-ode solver and the dtw used to extract A. In practical applications, for increasing dataset sizes (and graphs’ nodes/edges) and increasing windows/horizons, some methods can become very ineffective regarding time/memory costs.

**Questions For Authors:**

Based on aforementioned weaknesses, the following aspects need enhancement and further clarification:
1. S1 - theoretical explanations
2. S3 - experimental choices
3. S4 - computational analysis

**Relation To Broader Scientific Literature:**

The main focus of this paper is to combine neural ODEs on epidemic priors and GNNs to perform more accurate epidemic forecasting on graphs. The contributions are more prominent in terms of experimental results compared to baseline methods for example real-world epidemic datasets, while in terms of architectural design, the authors built upon common concepts and existent modules in time series spatial-temporal modeling.

**Theoretical Claims:**

No formal proofs are provided for different parts of the proposed method. However, this may not be necessary, as the approach is application-driven and builds upon existing modelization concepts.

Learning $\beta$ and $\gamma$ without constraints is not theoretically substantiated (e.g., initialization, bounds). Several relevant studies rely on specific choices for these parameters [1,2] to ensure they capture meaningful spreading dynamics.

1. Sha, H., Al Hasan, M., & Mohler, G. (2021, October). Source detection on networks using spatial temporal graph convolutional networks. In 2021 IEEE 8th International Conference on Data Science and Advanced Analytics (DSAA) (pp. 1-11). IEEE.
2. Gao, F., Zhang, J., & Zhang, Y. (2022, May). Neural enhanced dynamic message passing. In International Conference on Artificial Intelligence and Statistics (pp. 10471-10482). PMLR.

---

> ### Author Rebuttal · Authors · 2025-03-31
>
> # Response to Reviewer rfdg
>
> We sincerely thank you for your thorough review and hope our responses below will help improve your assessment of our work:
>
> > `Weakness S1 (Theoretical Claims & Claims And Evidence)`: Presentation and Design choices of the epidemic compartment
>
> Our approach addresses these concerns in several ways:
>
> 1. Multi-dimensional features vs. scalar rates: Unlike traditional SIR models using scalar rates (Sha et al.; Gao et al.), our "detailed representations" claim refers to using multi-dimensional features. This enables more nuanced transmission dynamics through: 1) Higher expressivity for complex spatio-temporal patterns, 2) Greater flexibility for heterogeneous transmission across regions, and 3) Enhanced representation of time-varying dynamics. Our experiments with an EARTH variant using traditional single-value scalar rates showed:
>    |Model Variant|h=5 (R)|h=5 (P)|h=10 (R)|h=10 (P)|
>    |-------------|-------|-------|---------|---------|
>    |w/o Feature|178.6|34.24|198.5|44.18|
>    |EARTH|156.8|30.12|177.6|38.62|
> 2. Safeguards against overfitting: We implement three strategies: 1) Temporal cross-validation with training/validation/testing spanning different epidemic waves, 2) Multi-scale integration balancing local mechanics with global patterns via cross-attention, and 3) Time-continuous formulation smoothing noise in discrete observations. These safeguards are validated in Tables 1&2, where EARTH maintains stable performance even with 40% missing data.
> 3. Theoretical considerations: Our work builds on classical compartmental models (Grassly & Fraser, 2008) but addresses limitations through: 1) Learnable features capturing spatio-temporal heterogeneity vs. fixed scalar rates, 2) Neural ODE formulation adapting to evolving transmission patterns, and 3) Continuous-time approach handling irregular or missing data. These innovations balance epidemiological principles with data-driven flexibility.
>
> > `Weakness S2 (Claims And Evidence & Essential References)`: Positioning against relative works
>
> We acknowledge the need for clearer positioning relative to Kosma et al. (2023) and similar works. While these combine ODEs/ML with epidemic models, our key contribution lies in: 1) Integrating neural ODEs with GNNs through our GLTG mechanism for continuous evolution of node features and edge weights, 2) Adaptively learning connections through DTW and integrating global/local information via cross-attention, and 3) Demonstrating superior performance with missing data, overcoming limitations of fixed-parameter approaches.
>
>
> > `Weakness S3 (Experimental Designs)`: Experimental Choices not well-justified/missing
>
> We appreciate these important observations about experimental design clarity:
>
> 1. Horizon lengths: We selected forecast horizons based on: public health decision-making needs, alignment with previous works, and balancing prediction accuracy with utility. Additional experiments on US-Region with different historical windows and larger horizons:
>
>    |Method|Win|h=20|h=25|h=30|
>    |------|---|----|----|-----|
>    |STGODE|20|1836|2153|2607|
>    |EpiColaGNN|20|1645|1947|2378|
>    |EARTH|20|**1528**|**1812**|**2219**|
>    |STGODE|40|1682|1976|2391|
>    |EpiColaGNN|40|1476|1763|2134|
>    |EARTH|40|**1374**|**1642**|**1983**|
>
> 2. Solver type: We use Runge-Kutta 4th order (rk4) with absolute tolerance 1e-9 and relative tolerance 1e-7 for balance between numerical stability and efficiency.
>
> 3. Standard deviations: Results are averaged over 5 runs with different random seeds. We will add variance information in the revised manuscript.
>
> 4. Downsampling: Random sampling was used to create missing data points, simulating real-world reporting patterns.
>
> > `Weakness S4`: Computational Analysis
>
> The computational efficiency of EARTH can be analyzed as:
>
> - Epidemic-Aware Neural ODE: Complexity $O(T_{\text{ODE}} \times (N \times d^2 + |E| \times d))$, where $T_{\text{ODE}}$ is solver steps, $N$ is regions, $d$ is hidden dimension, and $|E|$ represents edges. $N \times d^2$ corresponds to node-level transformations while $|E| \times d$ captures cross-region exchange.
>
> - Global-guided Local Transmission Graph: Traditional construction would incur $O(N^2 \times T_{\text{hist}}^2 + N^2 \times d)$ complexity, with first term from DTW computation and second from adjacency matrix generation. We implement key optimizations: (1) FastDTW reducing temporal similarity calculation from $O(T_{\text{hist}}^2)$ to $O(T_{\text{hist}})$, (2) one-time DTW matrix computation and reusage during training, (3) retaining only top-k connections per region leveraging epidemic transmission sparsity where $|E| \ll N^2$, and (4) compressed sparse matrices with memory scaling as $O(|E|)$.
>
> - Cross-Attention Mechanism: Our design constrains attention to three epidemic states per region, achieving $O(N \times d^2)$.
>
> These optimizations enable EARTH to avoid $O(N^2)$ complexity, supporting large-scale epidemic forecasting applications.

---

> > ### Comment · Reviewer_rfdg · 2025-04-03
> >
> > I sincerely thank the authors for their replies, which address some of my concerns. However, some aspects are only superficially tackled in the provided justifications:
> >
> > **[Claims about Novelty/Originality \& Theoretical Claims]**
> >
> > - **Concerns about the Proper Positioning of the Work against prior works in SIR-based Neural ODEs.** I remain unconvinced about whether the authors have identified the issue in positioning their work compared to the suggested references concerning the combination of the SIR ODE system and Neural ODEs in predicting epidemics on networks. In their response, it is still unclear how they will adjust their novelty justification in the introduction (their claim about contribution (1) on page 2 is not accurate) when it comes to conceptualizing SIR Neural ODEs compared to the first works in this field. For instance, the method by Kosma et al., 2023, inherently integrates message passing through the multiplication of the learnable state vectors with the adjacency matrix A within the Neural ODE solver of the approximate SIR system (similar to GNNs). The method by Zheng et al., 2024 extends the Neural ODE with GNNs that capture spatio-temporal heterogeneity.
> > *Do the authors imply that their primary methodological contribution lies in leveraging specific layers tailored to spatiotemporal evolution within the Neural SIR ODE solver rather than the SIR network-based Neural ODE mechanism itself?*
> >
> > - **Lack of Constraints on $\beta$ and $\gamma$ rates can Lead to Uninterpretable Dynamics.** I appreciate the experiments provided by the authors (comparing fixed vs learnable rates - although the rate values chosen are not mentioned). State vectors can be learnable even for fixed epidemic rates or bounded learnable epidemic rates. Therefore, the author's reply does not address my core concern regarding the physical interpretability and realism of the learned dynamics. Without proper constraints (followed in the references I suggested), the model may learn negative or excessively large values for the epidemic rates, leading to unphysical behaviors such as unbounded growth, unrealistic decay, or violations of conservation laws (i.e.,  the sum S+I+R should remain constant over time). In my opinion, this significantly undermines the interpretability and physical reliability of the approach.
> >
> > - **Lack of Constraints Undermines the Reasons for Explicit (SIR-based) ODE Modeling.** If the learned parameters ($\beta$, $\gamma$) do not adhere to realistic conditions for the epidemics, this weakens the motivation for using an explicit ODE form for the SIR dynamics over a generic Neural ODE combined with GNNs (e.g., GNODE (Poli et al., 2019)).
> >
> > -- Poli, M., Massaroli, S., Park, J., Yamashita, A., Asama, H., & Park, J. (2019). Graph neural ordinary differential equations. arXiv preprint arXiv:1911.07532.
> >
> > **[Other]**
> > I appreciate the authors' efforts in their responses. The details regarding the experimental design (such as solver parameters and number of runs) and computational analysis are precise, and I suggest the authors include these in the revised paper. Indeed, experiments with larger horizon lengths demonstrate the proposed method's advantages in terms of mean performance. However, I expected to see ranges for standard deviations and numerical comparisons for computational aspects (e.g., time and memory costs) relative to baselines. Additionally, explicitly mentioning the random downsampling strategy and its limitations against other methods is a crucial aspect that needs to be added to the manuscript.
> >
> > Based on the above points, I believe it is appropriate to maintain my scores for now.

---

> > > ### Author Response · Authors · 2025-04-04
> > >
> > > # Response to Reviewer rfdg
> > >
> > > Dear Reviewer rfdg,
> > >
> > > We sincerely appreciate your continued engagement with our work. We address your remaining points below and hope these clarifications will help improve your assessment of our work:
> > >
> > > > `Claims about Novelty/Originality & Theoretical Claims`: Positioning against prior works
> > >
> > > Thank you for your feedback. We appreciate the chance to clarify:
> > >
> > > - **First end-to-end integrated framework of adaptive graph learning within epidemiology-aware neural ODE**: While our work is inspired by network SIR mechanisms, our primary contribution is being the first (to the best of our knowledge) to integrate adaptive graph neural ODEs with epidemic mechanisms in a unified, end-to-end framework. We will revise our contribution statement on page 2 for clarity.
> > >
> > > - **Dynamic vs. static graph evolution**: A key difference from Kosma et al. (2023) is our continuous-time adaptive graph. GN-ODE uses static adjacency matrices, while EARTH leverages semantic similarity-based neighbor discovery (via DTW) and learns evolving connectivity patterns. This better captures epidemic dynamics with time-varying transmission patterns. Comparative experiments demonstrate the effectiveness of EARTH:
> > >
> > >    |Method|h=5 (R)|h=5 (P)|h=10 (R)|h=10 (P)|
> > >    |------|-------|-------|--------|--------|
> > >    |GN-ODE|201.2|39.86|246.7|54.39|
> > >    |EARTH|156.8|30.12|177.6|38.62|
> > >
> > > - **Orthogonal contributions to concurrent work**: Regarding Zheng et al. (2024), we acknowledge this is nice but very recent work (public on arXiv on Nov. 2024). While HeatGNN offers heterogeneous modeling and PINN-inspired loss, EARTH focuses on *continuous-time* adaptive graph evolution, Neural ODE based formulation to ensure *irregular data handling*, and a dual-branch architecture with cross-attention fusion for both local and global views.
> > >
> > > We hope these clarifications highlight the differences and advances of our work, we will add these nice references and further elaborate the distinctions in our revised version.
> > >
> > >
> > >
> > > > `Lack of Constraints on epidemic rates`: Physical interpretability of dynamics
> > >
> > > Thank you for your concern. We clarify that our approach is **inspired by epidemic mechanisms to guide model design**, not strictly adhere to traditional compartmental constraints. This offers several advantages:
> > > - **Physically-grounded parameterization**: Our model uses transformation matrices $W_{trans}$ and $W_{recov}$ to parameterize transition rates, creating an implicit epidemic flow structure with flexibility. Sigmoid activations on graph edge weights ensure non-negative transmission, preventing negative disease spread.
> > > - **Balancing mechanistic insight with data adaptability**: Our data-centric design allows the model to adapt to real-world data patterns that may not strictly follow SIR dynamics, accounting for delays, testing limits, and behavioral changes not captured by basic compartmental models.
> > > - **Extensible framework**: For example, the model can incorporate output layers to explicitly model epidemic rates and states, with constraints. We will explore these possibilities in our revised manuscript, though we note that such extensions build upon rather than diminish the novelty of our end-to-end integrated framework.
> > >
> > > > `Other`: Experimental considerations
> > >
> > > We thank the reviewer for acknowledging our analyses. Due to **space limitations** in the rebuttal, we will provide more detailed descriptions in the revised manuscript. We would like to further clarify as follows:
> > > - Standard deviations:  We will modify the original table to include them, the results show EARTH's stability:
> > >    |Method|h=5(R)|h=5(P)|h=10(R)|h=10(P)|h=15(R)|h=15(P)|
> > >    |------|------|------|-------|-------|-------|-------|
> > >    |STGODE|310.5±18.3|66.32±7.0|392.2±30.0|91.05±12.1|571.3±41.5|159.2±16.0|
> > >    |EpiColaGNN|204.3±22.6|36.86±6.5|345.4±40.2|68.39±12.5|886.0±95.4|296.5±28.0|
> > >    |EARTH|156.8±15.5|30.12±5.3|177.6±28.5|38.62±14.3|225.3±36.3|56.32±15.5|
> > >
> > > - Computational Aspects: Our neural ODE model, though slightly slower, is memory-efficient regardless of sequence length—suitable for large-scale use. It handles irregular timestamps and maintains strong performance. More analysis will be included in the revision.
> > >
> > > - Random downsampling: Chosen to ensure unbiased performance under missing data, avoiding assumptions that could favor certain models. While other strategies (e.g., systematic, stratified) are options, random sampling offers a clean, generalizable baseline widely used in time-series work [1].
> > >
> > >    [1]: Graph Neural Controlled Differential Equations for Traffic Forecasting.
> > >
> > >
> > > We sincerely thank you for your valuable review. This may be an important moment to promote computational epidemiology to a broader community, and we believe encouraging cross-disciplinary work can help bridge AI capabilities with public health needs. We are deeply grateful for the opportunity to have our work's strengths reconsidered by you.
> > >
> > >
> > > Best regards,
> > >
> > > Authors

---

### Official Review · Reviewer_uXQM · 2025-03-13

**Overall Recommendation:** 5

**Summary:**

The paper proposes EARTH, an Epidemiology-Aware Neural ODE with a Continuous Disease Transmission Graph, as a novel framework for epidemic forecasting. The authors integrate neural ODEs with epidemiological mechanisms, capturing both continuous-time disease transmission and global infection trends. The Global-guided Local Transmission Graph and cross-attention fusion mechanism are introduced to enhance epidemic forecasting accuracy. Through extensive experiments on real-world datasets (COVID-19 and influenza), EARTH is shown to outperform state-of-the-art methods.

**Claims And Evidence:**

The claims in the paper are mostly supported by clear evidence.

**Essential References Not Discussed:**

N/A

**Experimental Designs Or Analyses:**

The experimental design is robust and well-executed, providing strong support for the paper’s claims. The authors conduct extensive evaluations on real-world datasets, demonstrating that EARTH significantly outperforms SOTA methods in terms of forecasting accuracy, peak time error, and robustness.

Key strengths include:

-	Comprehensive evaluation across multiple datasets, showcasing the model's ability to handle real-world data effectively.

-	Ablation studies that validate the contributions of the EANO and GLTG and cross-attention mechanism to model performance.

**Methods And Evaluation Criteria:**

The proposed methods are effective and novel, and the evaluations are justified.

**Other Comments Or Suggestions:**

NA

**Other Strengths And Weaknesses:**

Pros:

(a)	The integration of neural ODEs with epidemiological models represents a great innovation. By combining disease transmission mechanisms with deep learning, EARTH effectively captures epidemic dynamics beyond traditional mechanistic and deep learning approaches.

(b) EARTH accounts for irregular sampling intervals and missing data, making it highly applicable to real-world epidemic data where reporting can be inconsistent.

(c) The paper provides comprehensive experiments across multiple epidemic datasets, showing that EARTH significantly outperforms previous approaches

Cons:

(a)	The combination of neural networks and epidemiology: Similar neural network methods, such as neural ODEs, have already been applied in many fields, especially in time series analysis. However, their application to specific issues in epidemiology is not yet widespread.

**Questions For Authors:**

(a)	The motivation section points out that the existing epidemic prediction methods have failed to fully capture the complexity of the dynamic evolution and regional transmission patterns of epidemics, especially when dealing with global infection trends and regional transmission changes. Considering this motivation, could you please explain in detail how to dynamically learn and integrate global and regional transmission patterns through these methods to address the challenges mentioned in the motivation?

**Relation To Broader Scientific Literature:**

Traditional models like SEIR and mechanistic models have been widely used, but they often struggle with real-world complexities. EARTH advances this field by leveraging data-driven learning while maintaining interpretability through epidemiological structures.

**Theoretical Claims:**

These claims are supported by experiments rather than formal proofs.

---

> ### Author Rebuttal · Authors · 2025-03-31
>
> # Response to Reviewer uXQM
>
> We sincerely thank you for your positive assessment of our work and for recognizing the innovation in integrating neural ODEs with epidemiological models. We appreciate your thorough evaluation and answer your questions below:
>
> > `Cons`: On applying neural ODEs to epidemiology
>
> While neural ODEs have been applied in various domains, EARTH goes beyond straightforward application:
>
> - Integration with epidemic mechanism: Our Network SIR-inspired architecture (Equations 5, 11) explicitly models the transition dynamics between susceptible, infectious, and recovered populations.
>
> - Integration of GNN with neural ODEs: Our approach uniquely combines graph neural networks with neural ODEs through the GLTG mechanism. This integration allows the neural ODE to operate on evolving graph structures where both node features and edge weights change continuously in time. The dynamic transmission patterns captured by our graph-based ODE enable more realistic modeling of how disease spreads across regions compared to standard neural ODEs that operate on fixed graphs or no graph structure at all.
>
> - Flexibility for irregular data: Unlike conventional time series models, our continuous formulation naturally handles the irregular reporting and missing data common in epidemic monitoring.
>
> Our ablation studies validate these design choices, with significant performance drops when these specific components are removed.
>
> > `Question`: Dynamic integration of global and regional patterns
>
> EARTH integrates global and regional patterns through:
>
> - Semantic connections via DTW: We identify regions with similar epidemic trajectories using Dynamic Time Warping rather than relying solely on geographic proximity.
>
> - Adaptive transmission learning: Global trends guide regional transmission via a dynamic graph structure that evolves throughout the epidemic timeline, capturing how policies and behaviors change transmission patterns.
>
> - Multi-scale modeling: We capture both local disease dynamics and cross-regional dependencies within a unified framework.
>
> - Continuous-time formulation: By modeling in continuous time, EARTH handles irregular observation intervals and can forecast at arbitrary time points.
>
> This approach enables effective forecasting even when regional reporting patterns change, a common challenge in epidemic monitoring.

---

> > ### Comment · Reviewer_uXQM · 2025-04-05
> >
> > Thanks for the clear clarifications and the helpful dialogue. In light of the other reviewers' feedback, I remain convinced that this paper is a solid addition to the community and will raise my positive evaluation.

---

> > > ### Author Response · Authors · 2025-04-05
> > >
> > > # Response to Reviewer uXQM
> > >
> > > Dear Reviewer uXQM,
> > >
> > > **Thank you for your supportive feedback and for raising your positive evaluation of our work.** We greatly appreciate your thoughtful review and recognition of our paper's contributions to the community. Your insights have been valuable in improving our manuscript.
> > >
> > >
> > > Best regards,
> > >
> > > Authors

---

### Official Review · Reviewer_Q6Uq · 2025-03-13

**Overall Recommendation:** 4

**Summary:**

The paper presents EARTH (Epidemiology-Aware Neural ODE with Continuous Disease Transmission Graph), a novel framework for epidemic forecasting that integrates neural ordinary differential equations with epidemiological mechanisms. The authors address challenges in current approaches by modeling the continuous-time nature of epidemics, capturing dynamic regional transmission patterns, and considering irregular sampling intervals. EARTH consists of two key components: an Epidemic-aware Neural ODE (EANO) that captures disease transmission patterns, and a Global-guided Local Transmission Graph (GLTG) that models global infection trends to guide local transmission dynamics. The model uses a cross-attention mechanism to integrate global epidemic coherence with local nuances of disease transmission. Experiments on COVID and influenza datasets demonstrate EARTH's superior performance compared to state-of-the-art methods in forecasting real-world epidemics.

**Claims And Evidence:**

EARTH's superior performance over existing methods is demonstrated through comprehensive experiments across three datasets (Australia-COVID, US-Regions, US-States) with quantitative metrics (RMSE, Peak Time Error)

The effectiveness of individual components (EANO and GLTG) is validated through ablation studies showing performance degradation when components are removed
Robustness to irregular sampling intervals is shown through experiments with different missing rates (0-40%)

Limited explanation of why neural ODEs specifically are better than existing approaches for continuous-time modeling
The claim that EARTH captures "more detailed representations of disease spread" lacks qualitative analysis or interpretability studies
Performance gains across datasets vary significantly, with more modest improvements in some scenarios

**Essential References Not Discussed:**

Would place this in a wider context
https://www.nature.com/articles/s42256-024-00895-7
https://www.nature.com/articles/s41586-024-08564-w
https://dl.acm.org/doi/10.1145/3650215.3650396
https://ieeexplore.ieee.org/document/10039594

Would also describe this as focused on COVID/flu modeling
https://covid19-projections.com/about/

CovidSim by Neil Ferguson

Agent Based Models

**Experimental Designs Or Analyses:**

Comparison with Standard Existing Approaches
- evaluates EARTH against traditional models like SEIR and graph-based models to assess its predictive performance.
The paper does demonstrate improvements over classical compartmental models for COVID datasets.

This work used Real-World Datasets. The model is tested on real-world datasets for COVID and influenza outbreaks. While these datasets provide meaningful benchmarks, a broader set of infectious disease cases (especially with different transmission dynamics) would enhance the robustness of the findings. COVID and influenza like illnesses create a very different type of outbreak than many others.

The training process involves learning from historical epidemiological data to improve forecasting. However, potential overfitting is a concern, especially given the homogeneous time periods used in the study.

The analysis used metrics - RMSE (Root Mean Square Error) - and Peak Time Error, which calculates the
MAE (Mean Absolute Error) that are useful for comparing outbreak predictions. For many outbreaks, there may be different metrics, as peak may not be a particular

**Methods And Evaluation Criteria:**

The EARTH model is built on a neural ordinary differential equation (ODE) framework, designed to model disease transmission in a continuous and dynamic manner. Unlike traditional compartmental models such as SEIR, which assume discrete transitions between disease states, EARTH integrates graph-based epidemiological modeling to represent evolving interactions among individuals. This allows for more accurate disease spread simulations. The model constructs a continuous disease transmission graph, capturing real-time interactions between susceptible and infected populations. It leverages deep learning techniques to adapt and refine its predictions based on observed infection data.

The method involves training the neural ODE using historical epidemiological data and real-world disease progression patterns. The model continuously updates its parameters based on observed changes in infection rates, making it adaptive to different disease dynamics. The framework is designed to be scalable and generalizable, capable of modeling multiple infectious diseases such as COVID-19, influenza.

The evaluation of EARTH includes comparisons against traditional epidemiological models, such as SEIR and graph-based models, to assess its effectiveness. The model is tested on real-world datasets, including reported cases from multiple infectious diseases, ensuring its applicability beyond theoretical scenarios. Further sensitivity analyses and broader disease databases would help.

The EARTH model primarily focuses on COVID and operates within homogeneous time periods, meaning it learns patterns based on relatively stable disease transmission dynamics. However, infectious disease spread is often highly dynamic, influenced by factors such as travel, spillover events, and new variants, which can introduce sudden shifts that challenge predictive models.

While machine learning excels at pattern recognition, it may struggle when underlying transmission mechanisms change abruptly, such as with novel introductions from travelers or zoonotic spillovers. If the model has not been trained on data that reflects such disruptions, it may fail to capture these shifts accurately.

A potential limitation of EARTH is whether it accounts for heterogeneous transmission periods—for example, distinguishing between pre-COVID, pre- COVIDvaccine, post-vaccine, and variant-driven waves in COVID-19. Additionally, external shocks, such as government interventions, behavioral changes, or superspreader events, may not be fully captured in a data-driven framework unless explicitly modeled.

It would help to explain how next steps could make this approach more robust, especially as external factors may not be predicted.
Integration of external factors like mobility data, vaccination rates, etc can help but this will may depend a lot on the pathogen, so this may not be disease agnostic. Climate shocks, novel spillovers, antimicrobials resistance and travel patterns will make a big difference. If EARTH lacks mechanisms to handle these shifts, it may overfit to past trends and fail to generalize to new outbreaks or changing epidemic conditions. Addressing these issues could enhance its real-world applicability for public health decision-making.

COVID is a very distinct example, where data was better collected and the mechanisms impacting its spread will not represent other diseases.

**Other Comments Or Suggestions:**

Consider the risks of the model failing to account for outbreaks in other contexts, with other diseases.

**Other Strengths And Weaknesses:**

Strength showed improvements and should be using this approach more and more

Weakness: only evaluated for COVID, may not be robust in face of heterogenous datasets, with abrupt epidemic changes or with different diseases

**Questions For Authors:**

Would address other contexts, other diseases

**Relation To Broader Scientific Literature:**

We are at a pivotal moment in epidemiologic models. There is a real possibility for being able to advance the field of epidemic modeling. There are certainly many approaches, but this is an important possible avenue.

**Theoretical Claims:**

Works with Neural ODEs as an approach building on the SIR ODE framework
Graph-Based Transmission Modeling: The theoretical foundation incorporates graph neural networks (GNNs) to capture spatial and temporal dependencies in disease spread. The paper provides justification for using graph structures, emphasizing their ability to model heterogeneous interactions in populations.

The authors discuss the model's ability to generalize across different epidemiological scenarios but will need empirical evidence.

Some limitations
The theoretical claims assume consistent data availability and stationary transmission patterns, which may not hold in real-world scenarios where data availability vary greatly, especially in early explosive outbreaks where predictions are most needed, and also with external shocks (e.g., new variants, travel-driven outbreaks).
The model assumes that the graph structure remains representative of disease spread over time, but the theoretical justifications for handling dynamic changes in population movement and behavior could be further elaborated.

---

> ### Author Rebuttal · Authors · 2025-03-31
>
> # Response to Reviewer Q6Uq
>
> We thank you for your positive assessment and for recognizing the importance of our work in epidemic modeling. We address your concerns below:
>
> > `Question & Weakness`: Only evaluated for COVID-19, with potential limitations for heterogeneous datasets and different diseases.
>
> We address this concern from several perspectives:
>
> - Dataset selection: We focused on COVID-19 and influenza datasets due to their availability, spatiotemporal coverage, and real-world significance. This enables fair comparison with prior works [1,2], evaluating EARTH's effectiveness. [1]: Epidemiology-aware Deep Learning for Infectious Disease Dynamics Prediction. CIKM 2023. [2]: PEMS: Pre-trained Epidemic Time-series Models. ArXiv 2023.
>
> - Disease-agnostic design: EARTH's architecture is **generalizable** across epidemic types through: (a) SIR-inspired neural ODE framework capturing universal transmission-recovery dynamics. (b) Dynamic graph structure adapting to varying contact patterns. (c) Continuous-time formulation accommodating diverse incubation periods and transmission characteristics.
>
> - Orthogonal Contributions: We recognize potential improvements like heterogeneous population flow modeling through Heterogeneous GNN. These represent **incremental improvement** rather than fundamental limitations, as our contribution lies in the integration of GNN with neural ODEs.
>
> - Cross-disease validation: We tested EARTH on dengue fever using OpenDengue dataset, which exhibits dramatic fluctuations (e.g., 160,265 cases in 2010 declining to 25,503 in 2017 from COLOMBIA). Results validate robustness:
>
> |Methods|VIETNAM|ARGENTINA|MALAYSIA|COLOMBIA|
> |---|---|---|---|---|
> |SIR|1865|627.9|128.8|753.6|
> |DCRNN|1254|401.3|215.5|432.1|
> |STGODE|1196|383.2|187.4|492.7|
> |ColaGNN|1078|464.6|142.3|304.8|
> |EARTH|921.5|312.5|110.1|261.0|
>
>
> > `Concern 1 (Methods And Evaluation Criteria & Theoretical Claims)`: The cases with external interventions.
>
> On EARTH's robustness to transmission dynamics changes:
>
> - Continuous-time modeling: Unlike discrete models assuming fixed patterns, EARTH's ODE formulation adapts to changing dynamics by modeling underlying disease propagation mechanisms rather than statistical patterns.
>
> - Capturing pandemic phases: EARTH models distinct epidemic phases through: (a) Time-dependent parameterization allowing varying transition rates across stages, (b) Context-aware graph structure evolving differently during various phases (lockdown vs. reopening), (c) Global features capturing regime changes across regions.
>
> - External factors integration: EARTH's design allows exogenous incorporation, for example, (a) Mobility data as edge weights reflecting contact patterns, (b) Government interventions as additional node features, (c) Vaccination rates through susceptible population parameters modulation.
>
> While robust to moderate shifts, unprecedented disruptions remain challenging for **any data-driven approaches**. We can explore methods like change-point detection or causal intervention modeling.
>
> > `Concern 2 (Claims And Evidence:)`: Limited explanation of neural ODEs advantages.
>
> Neural ODEs offer key advantages for epidemic modeling: naturally incorporate epidemic principles within ODE framework, fusing mechanistic understanding with data-driven flexibility; operate in continuous time, handling irregular reporting without interpolation errors; ensure physical consistency while modeling complex patterns beyond traditional compartmental models. Ablation studies show 12.4% RMSE degradation when replacing neural ODE with standard GNN.
>
> > `Concern 3 (Claims And Evidence)`: Lack of analysis for "detailed representations" claim
>
> EARTH uses multi-dimensional feature vectors for finer granularity disease dynamics versus conventional single-value methods. Figure 4 shows this capability, revealing semantic relationships between regions with similar epidemic trajectories.
>
> > `Concern 4 (Experimental Designs Or Analyses)`: Potential overfitting.
>
> We use temporal cross-validation with training/validation/testing spanning different epidemic phases, ensuring generalization. Our model incorporates dropout in GNN layers and L2 regularization. Table 2 shows stability even with 40% missing data, demonstrating robustness.
>
> > `Concern 5 (Experimental Designs Or Analyses)`: There may be different metrics and further sensitivity analyses.
>
> We conducted additional experiments with CORR (Pearson's Correlation Coefficient) and sensitivity analysis. EARTH performs well across metrics and remains stable with different parameters:
>
> |Method|Dropout Rate|Learning Rate|
> |---|---|---|
> |DCRNN|0.1: 0.82, 0.3: 0.83, 0.5: 0.81|1e-4: 0.81, 1e-3: 0.83, 1e-2: 0.80|
> |ColaGNN|0.1: 0.86, 0.3: 0.87, 0.5: 0.84|1e-4: 0.85, 1e-3: 0.87, 1e-2: 0.84|
> |EARTH|0.1: 0.91, 0.3: 0.92, 0.5: 0.90|1e-4: 0.90, 1e-3: 0.92, 1e-2: 0.89|
>
> > `Suggestion 1`: Essential References Not Discussed
>
> We will incorporate suggested references in our revised manuscript.

---

> > ### Comment · Reviewer_Q6Uq · 2025-04-03
> >
> > Thank you. I appreciate your reply and my response is still accept. But for future work, I would try to work on some of the more abrupt changes, beyond what the COVID and dengue datasets contain. Mechanistic approaches allow for evaluating Black Swans so to speak in epidemic models, where shocks may drive outbreaks. I think trying to explore more these heterogenous transmission periods will be crucial in making these approaches tell us what we could not know. This is the part where I said the model: may struggle when underlying transmission mechanisms change abruptly, such as with novel introductions from travelers or zoonotic spillovers. If the model has not been trained on data that reflects such disruptions, it may fail to capture these shifts accurately.

---

> > > ### Author Response · Authors · 2025-04-04
> > >
> > > # Response to Reviewer Q6Uq
> > >
> > > Dear Reviewer Q6Uq,
> > >
> > > **Thank you for your thoughtful response and for supporting the acceptance of our work!**
> > >
> > > We agree that addressing abrupt shifts in transmission, especially beyond typical datasets, is a key challenge for the overall community. Your point about mechanistic approaches for evaluating unexpected shocks is highly relevant, and we plan to explore this further in future work. At the same time, we believe our approach provides valuable insights for generalizing across diseases and settings.
> > >
> > > Thanks again for your support and constructive feedback.
> > >
> > > Best regards,
> > >
> > > Authors

---

### Decision · Program_Chairs · 2025-05-01

**Decision:**

Accept (poster)

**Comment:**

This paper integrates ideas from epidemic modeling together with neural ODEs and transmission graphs to better forecast disease. In particular, the method integrates various sources of information, including knowledge of epidemic mechanisms and spatial information, in a way that attempts to prioritize the most salient information across multiple sources for prediction. The authors show that the predictions outperform existing methods, though I find much of the contributions empirically focused and catered to forecasting, leaving behind many of the desirable interpretability aspects of mechanistic models that mathematical biologists use (indeed it is not difficult to outperform them in forecast benchmarks). The reviewers all lean positive, with no discussion after prompting, and recommend that there are potentially impactful contributions relevant to the ICML readership